# Emergence and retention of a collective memory in cockroaches

**Mariano Calvo Martín**[1,2]*, **Esméralda Rodriguez Palacio**[2], **Jean-Louis Deneubourg**[1], **Stamatios C. Nicolis**[1]

1 Center for Nonlinear Phenomena and Complex Systems (CENOLI), Université Libre de Bruxelles, Brussels, Belgium, 2 Unit of Evolutionary Biology and Ecology, Université Libre de Bruxelles, Brussels, Belgium

* mcalvoma@ulb.be

**Data Availability Statement:** Data are available from https://doi.org/10.6084/m9.figshare.21130477

## Abstract

The stability of collective decisions-making in social systems is crucial as it can lead to counterintuitive phenomena such as collective memories, where an initial choice is challenged by environmental changes. Many social species face the challenge to perform collective decisions under variable conditions. In this study, we focused on situations where isolated individuals and groups of the American cockroach (*Periplaneta americana*) had to choose between two shelters with different luminosities that were inverted during the experiment. The darker shelter was initially preferred, but only groups that reached a consensus within that shelter maintain their choice after the light inversion, while isolated individuals and small groups lacked site fidelity. Our mathematical model, incorporating deterministic and probabilistic elements, sheds light on the significance interactions and their stochasticity in the emergence and retention of a collective memory.

## Introduction

Aggregation is a widespread phenomenon encountered in many natural and artificial systems. It that emerges from the interactions between the constituting units and their responses to the environmental heterogeneities [1–4]. This phenomenon is involved in many spatiotemporal organizations and activities, from stress reduction in pre-socials species [5–7] to complex structures in eusocial animals [8, 9]. In these social systems, the individual responses to the environment are based on the intrinsic state of the animals [10] and can be drastically modulated by social interactions (e.g. conspecific presence). For example, it has been shown that shelter size preference of the spiny lobster (*Panulirus argus*) shifts from shelters scaled to their body size to collective housing in large shelters when the conspecific density increases [11]. Similarly, individual response of the American cockroach (*Periplaneta americana*) for a vanillin scented shelter is inverted in the presence of conspecifics [12].

Among the potential mechanisms generating these different density-dependent responses, many studies have highlighted the importance of inter-attractions in generating positive feedbacks that compete with each other and their role on the observed patterns [13, 14]. In these situations, quantitatively identical networks of feedbacks (positive and

**Funding:** This research was realised without any funding.

**Competing interests:** The authors have declared that no competing interests exist.

negative) can lead to different collective decisions such as the selection of the better beneficial option, or a sub-optimal choice (e.g. exploiting a less nutritive food resource) in which the group can be trapped [15, 16].

The potential simultaneous presence of several collective responses and their stability (multistability) depend on the interplay between the strength of the feedbacks and the difference between the characteristics of the options [17, 18]. The dynamics and the frequency of the different choices are affected by the stochasticity of the individual behaviour and the initial distribution of the population (initial conditions). In a constant environment, shelter selection in gregarious animals [19, 20] usually leads the group to choose one shelter, which stays occupied over time (stable stationary state) despite some punctual perturbations or activities (e.g. periodical) [21]. On the other hand, in a natural environment subjected to changes (e.g. degradation of a site), depending on the properties of the system and on its feedbacks' network, two very different outcomes are observed. If the adopted outcome only depends on the parameters, whatever the level stochasticity is, the system is monostable [22, 23]. In multistable systems, the settlement in a particular state (the outcome) does not depend only on the new environmental parameters but also on the state it has been settled previously. Such systems are able to exhibit an hysteretic behaviour, a form of memory which is not necessarily encoded at the individual level and which emerges through the network of feedbacks present. This behaviour has been previously described in artificial systems [24, 25], in ecosystems [26, 27], in biomolecular systems [28] and in social systems such as spiders [29] or ants [30].

A deterministic description can however be misleading because it ignores the stochasticity of the processes which is particularly crucial when the number of individuals is small. In this case, the stochastic effects can lead to spontaneous transitions from one state to another and therefore to bimodal (or even multimodal) distributions. Of special interest is the transient bimodality occurring during the shift from an initially single stable state to another one after an environmental change [31, 32]. The question arises therefore, whether transient or stationary bimodalities can be differentiated experimentally since stationarity is rarely observed in the time scale of most biological systems.

In social systems (i.e. involving interactions), the memory is referred to as a collective memory when shared and displayed by the group, regardless the existence or not of a multistability or hysteretic behaviour [33–37]. In many cases, collective memory is related to cultural transmission in which the information stored at the individual level is propagated/maintained through social interactions [38–41]. Moreover, external factors can also participate to the storage of the memory such as trail in ants in which the surrounding environment is modified [42, 43]. In this paper, we show that collective memory can also emerge solely through the interactions between the individuals and can be stored in the social structure.

From a theoretical point of view, collective memory is based on the nonlinear dynamics generated by social interactions that can in some cases lead to multistable patterns [44]. These properties have been shown to be present in gregarious species [19, 45, 46] such as the American cockroach [47]. In this context, a previous study [21] has shown the capacity of this insect to maintain a collective decision when faced to short environmental perturbations. In this study, we explore the possibility of such stability after a permanent change of the environment, which would be a signature of the existence of a collective memory.

Thanks to a synergy between experiments and mathematical modelling, we highlight the emergence of a collective memory phenomenon during sheltering dynamics of groups of *P. americana* submitted to a permanent light inversion between two shelters. The multistable pattern reported here, where groups collectively select the dark or the light shelter at the end of the trials, is the result of groups expressing a collective memory. More specifically, groups having reached a large aggregate (consensus) in the darker shelter before the light inversion show

a fidelity to this shelter even though the shelter has become unfavourable. On the other hand, isolated individuals and small aggregates do not express any fidelity to the initial site. Our experimental and theoretical results highlight the crucial roles played by the number of interactions among individuals and their stochastic processes in the emergence and retentions of a collective memory.

## Materials and methods

### Biological model

Cockroaches of the species *P. americana* are reared in a room at 25° C under a 12 h:12 h light: dark cycle and a relative humidity of 40%. The insects are bred in boxes of circa 1000 individuals in all stages of development and with both sexes mixed. The cockroaches have access to water and dog food pellets (*Tom & Co*Ⓡ) ad libitum. 24 hours prior to the experiments, isolated individuals and groups of 10 nymphs of L6-L8 instar (at these larval stage the sex of the cockroach is non recognizable at naked eye) are transferred in a box (15.5 × 11 × 6 cm) with water and food. Cockroaches nymphs have no sexual behaviours, which allow them to better respond to conspecifics and form aggregates [48]. Cockroaches did not have access to water or food during the experiments.

### Experimental set-up

The device (Fig 1 and S1 Fig in S1 File) is a Plexiglas box (L = 35 cm, W = 23.5 cm, H = 13.4 cm) in which two circular shelters (diameter = 9.5 cm, H = 1.2 cm) with lateral opening (W = 2 cm, H = 1.2 cm de hauteur), one characterized by a red light (630 nm at 100 lux) and the other one by a green light (560 nm at 100 lux). Cockroaches, as numerous arthropods perceive poorly the red light spectrum and on the opposite they perceive quite well the green light spectrum [49]. A circular arena of 12.5 cm of diameter connects the shelters through their opening (Fig 1) and is characterized by a white light at 1000 lux. The different light zones (shelters and circular arena) are achieved by dimmable LED bands (APA102, 30 LED/m, DC 5V, white stripe), controlled by a script through a microcontroller board. Each light zone is separated by a white carboard board to reduce external lights/shades perturbations. The bottom of the experimental set-up is covered with a white paper layer covering the floor of the shelters and the arena which is replaced after each trial to avoid floor hydrocarbon marking [50].

### Experimental procedure

Isolated individuals (N = 26) or groups of 10 cockroaches (N = 27), without any external damage, are faced to the choice between two shelters under two different conditions.

1. The control condition: isolated individuals (N = 5) and groups (N = 5) explore the experimental set-up in which the luminosity of each shelter is constant through the entire length of trial (1320 minutes) (Fig 2). One shelter is illuminated with a red light (hereafter the R shelter) and the other one with a green light (hereafter the G shelter).

2. The inversion condition: isolated individual (N = 21) and groups (N = 22) explore the same experimental set-up than the control condition, but after 600 minutes, the light characteristics of the shelters are gradually inverted. This process lasts 60 minutes and the conditions remain constant until the end ($T_{end}$ = 1320 minutes) (Fig 2). From 600 to 660 minutes, the R shelter shifts therefore from red to green light (hereafter the RG shelter), while the G shelter is shifting from green to red light (hereafter the GR shelter). This inversion of luminosity is controlled by the microcontroller board.

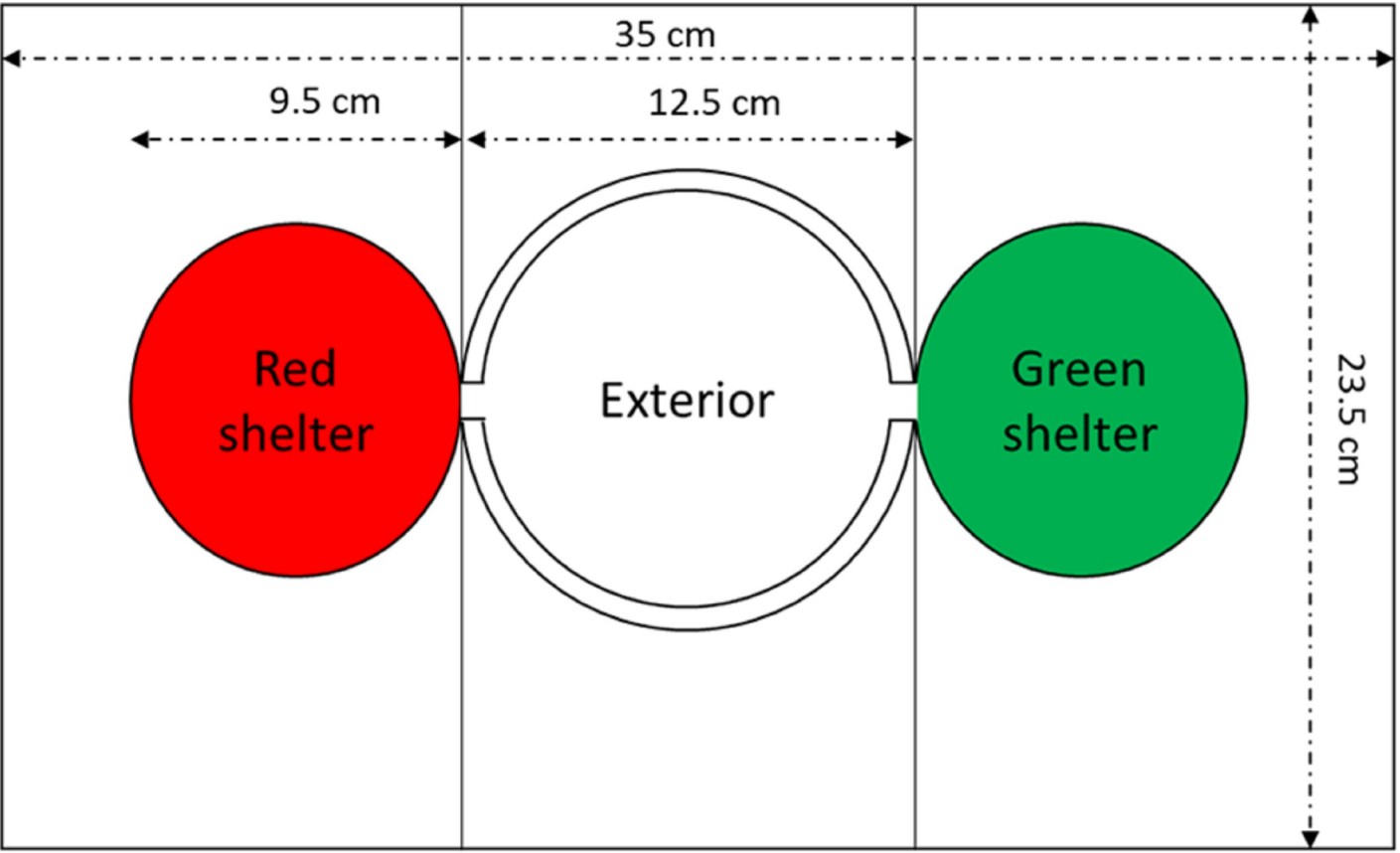

**Fig 1. Sketch of the experimental set-up (not scaled).**

## Data and statistical analysis

We recorded by a video camera (Logitech C920), located over the set-up (S2 Fig Material and Methods section in S1 File), at 17 frames per second for the entire duration of the trials. The number of individuals in each shelter is recorded every time-step (30 minutes).

Data and statistical analysis are performed using R studio (R Core Team, 2018, R Foundation for Statistical Computing, https://www.r-project.org/). The significance of the statistical tests is fixed to $\alpha = 0.05$. We used linear regressions to analyse the global sheltering process. The selection of shelters is a dynamic process that is typically nonlinear and/or exhibits varying variance over time, particularly in gregarious and social insects. We used three methods of permutation and resampling tests. (1) The complete combination method, which gives all the combinations of the sheltered population of a sample of trials taken from a larger set of trials (eq. S1 material and methods section in S1 File); (2) the complete permutation method, which gives all the permutation of the population between the shelters for a set of trials (eq. S2 material and methods section in S1 File); and (3) the random permutations, for which in each trial of a set, the population between the shelters is randomly permuted ($10^7$ iterations). These permutations and sampling tests are used to compare the distribution of individuals in each shelter at every time-step, as well as the cumulated presence of individuals in each shelter for a specific time period. This allows to do intra/inter-conditions and intra/inter-population

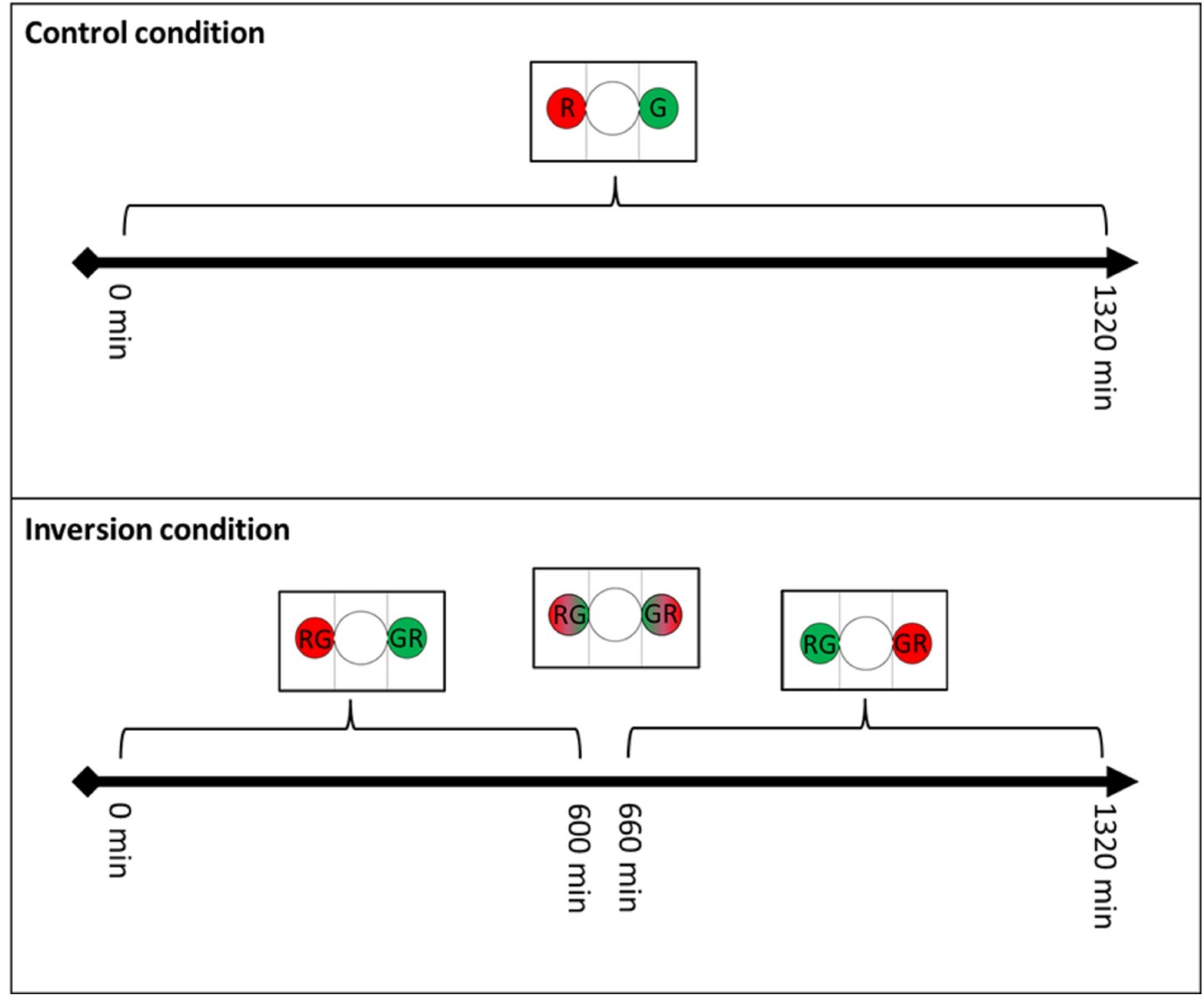

**Fig 2. Temporal diagram of the shelter luminosities depending on the experimental condition.**

comparations. Furthermore, the difference between the proportion of the total population each shelter is analysed using the Cramer-Von Mises test. Spearman correlation test and $\chi^2$ test or Fisher exact test is used to analyse the relationship between the observed presence before and after the shelter luminosities inversion. In the sequel, we define a consensual selection of a shelter as a selection where the majority of the total sheltered population is significantly different than expected from a binomial distribution. Finally, we used goodness of fit tests to compare experimental and theorical results.

## Results

### Global sheltering process

The sheltering process in cockroaches is governed by their photonegative behaviour [51]. The total sheltered population, calculated as the sum of the number of individuals in each shelter and the proportion being this total, divided by the total population in a trial: $\frac{Total\ sheltered\ population}{Total\ population}$. This proportion rapidly increases both for the isolated individuals and the groups (Fig 3). At 30 minutes, the mean ± SEM is 0.73± 0.1 for isolated individuals (pooled isolated individuals: N = 26 = $N_{Control}$ + $N_{inversion}$ = 5 + 21) and 0.89 ± 0.004 for groups (pooled Groups: N = 27 = $N_{Control}$ + $N_{inversion}$ = 5 + 22). Between 30 and 1320 minutes, there is no significant influence of the time on the sheltered proportion for isolated individuals (N = 26; Linear regression: $R^2$ = 0.0017, F = 2.07, P = 0.15 –Intercept = 0.68, t = 25.601, P $<$ 2x10$^{-16}$ –Slope = 5x10$^{-5}$ minute$^{-1}$, t = 1.43, P = 0.15, see S3 Fig in S1 File). Regarding the groups, the influence is significant but weak (less than 0.4 individuals in 22 h), as indicated by the slope value (N = 27; Linear regression: $R^2$ = 0.008, F = 11.35, P $<$0.001 –Intercept = 0.88, t = 132.9, P $<$ 2x10$^{-16}$ –Slope = 2.9x10$^{-5}$ minute$^{-1}$, t = 3.36, P $<$ 0.001, S3 Fig). The positive influence of conspecifics on the sheltering process is highlighted by the larger value of the intercept in the groups and the non-overlapping confidence intervals (isolated individuals: 0.63–0.73; groups: 0.87–0.89). Furthermore, the inversion of light after 600 minutes has no influence on the proportion of the total sheltered population neither for the isolated individuals and the groups (see Global dynamics section, S3 Fig in S1 File).

### Shelter colour influences

For the first 600 minutes, no significant difference is observed between the control and the inversion conditions for isolated individuals and groups (see Table 1 and Shelter colour influence section in S1 File). Therefore, we pooled these conditions for each population size. For the pooled isolated individuals, the cumulative presence inside the red shelter, between the time-steps 0 and 600 minutes (mean ± SEM: 8.27 ± 0.93 individual) is significantly higher than the cumulative presence inside the green shelter (5.46 ± 0.78) (Table 1). However, at every time-step no difference is observed (except at one time-step, Table 1). This is in agreement with cockroaches colour vision, as many other arthropods do not perceive (or not well) the red-light wavelengths (from 600 nm to 700 nm) [52]. Thus, the red-light illuminated shelter is perceived as the darkest one, and the one illuminated with a green light is only darker than the exterior. As for the pooled groups, the cumulative presences inside the red shelter after 600 minutes (134.63 ± 5.11) and at every time-step (44.18 ± 3.45) is significantly larger than the one in the green shelter (see Table 1). Individuals discrimination between the two shelters increased also with the population size, in agreement with a common feature in social animals where public information shared through interactions amplifies individual discrimination [20, 53, 54].

Previous studies have shown that the interactions between individuals stabilize the choice as compared to an isolated individual and can lead to multistability, meaning that consensuses are reached for either the preferred or the less preferred shelter depending on the initial conditions and on the stochasticity [20, 53]. This suggests that such dynamical processes become irreversible once a particular number of individuals (threshold) is aggregated in one shelter [55–57]. In our experiments, and in particular, in the control conditions, groups reached a more stable selection than the isolated individuals (Table 1 and S4 Fig in S1 File). This stability of the group allows its members for multiple cooperative activities/benefits [58, 59].

The stability of the decision is mainly expected to occur when the environmental qualities (abiotic and biotic) do not change over time, as in the control condition. However, in most

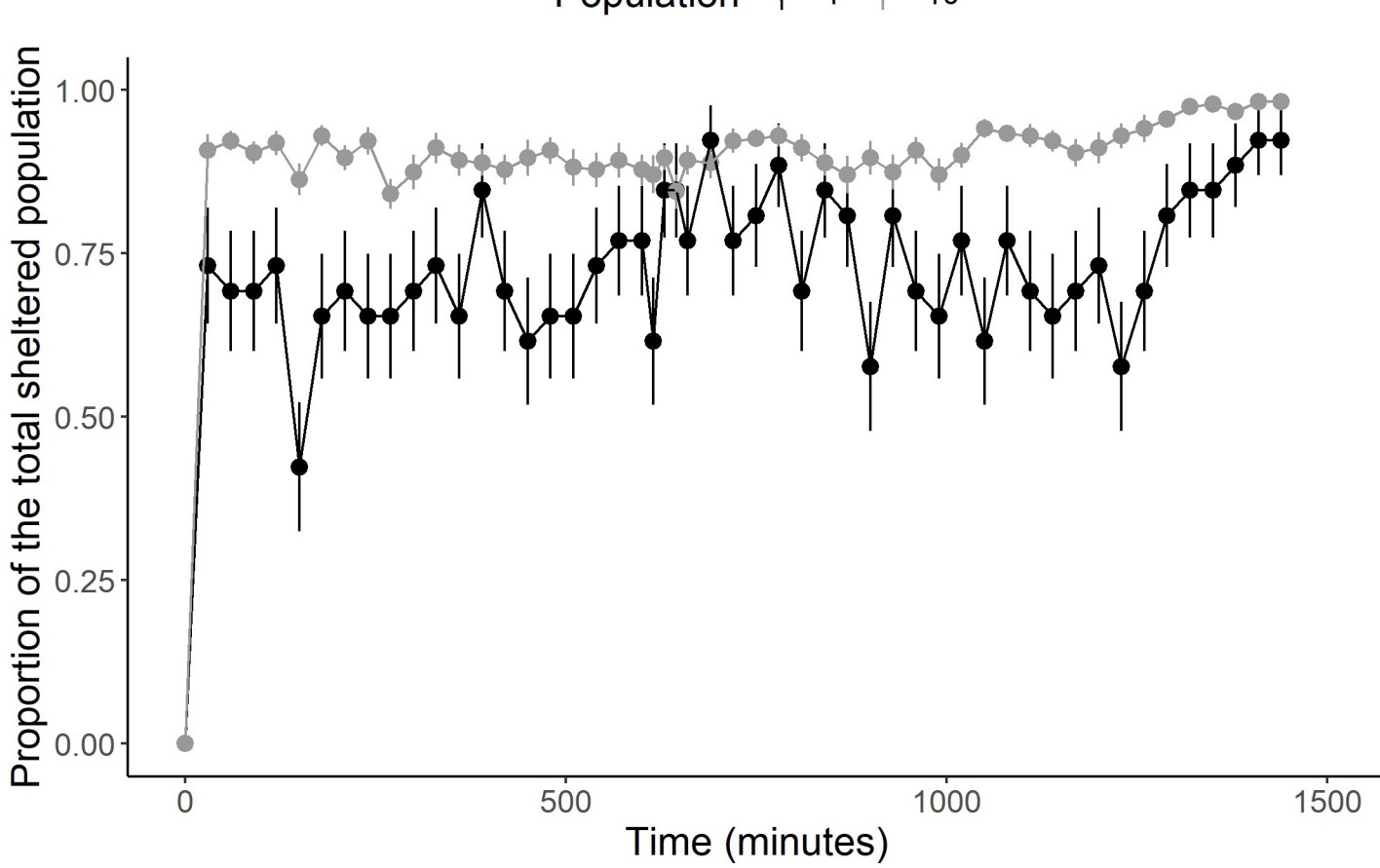

**Fig 3. Sheltering process of the total population over time.** Mean ± SEM of pooled isolated individuals (black; N = 26) and of pooled groups (grey; N = 27).

natural systems, static environments are rarely observed. If the quality of a resting site decreases, the decision to keep staying in the site is counterintuitive., as it implies a potential decrease of fitness for individuals (isolated or in groups). However, theoretical studies involving multistable systems predict that under some conditions the initial state is robust and stable after a change of the parameters [21, 60, 61]. The purpose of the inversion condition experiment is to demonstrate this stability. The results show that the cumulative presence of isolated individuals between 660 and 1320 minutes (mean ± SEM) differs significantly (Table 1) between the new red shelter (12.33 ± 0.87; GR shelter) than the new green shelter (4.38 ± 0.36, RG shelter). This difference becomes less pronounced at each time-step, with a significant difference of the presence of individuals between shelters for 12 out of 22 time-steps. However, for the groups, the cumulative presences in the GR and RG shelters does not show a significantly difference (respectively: 129.5 ± 16.58 and 82.95 ± 17.69, Table 1). Regarding the time-steps, no significant difference is observed between the shelters, except at time 1320 minutes (Table 1; see also S4–S6 Figs in S1 File Population influence section). These results strongly suggest that the light inversion impacts the distribution of individuals between the shelters in a social context.

**Table 1. Permutation tests.**

| Compared conditions | Interval time (min) | Resampling method | Permutation test method | P value summary for each time-step | P value for cumulative |
|---|---|---|---|---|---|
| Iso. cont. vs iso. inv. | 0 to 600 | Iso. cont.: 5 vs Iso. inv.: 5 of 21 | Complete combination N = 20349 | P > 0.05 Except at times: 330, 360, 720 minutes | P = 0.12 |
| Iso. pooled Red sh. vs Green sh. | 0 to 600 | Iso pooled: 26 | Complete permutation N = 67108864 | P > 0.05 Except at time: 180 minutes | P = 0.037 |
| Iso. cont. vs iso. inv. | 660 to 1320 | Iso. cont.: 5 vs Iso. inv.: 5 of 21 | Complete combination N = 20349 | P > 0.05 Except at time: 810, 840, 1260 minutes | P < 0.61 |
| Iso. pooled Red sh. vs Green sh. | 660 to 1320 | Iso pooled: 26 | Complete permutation N = 67108864 | P < 0.05 Except at time: 660, 720, 810, 1080–1140, 1200–1320 minutes | P < 0.0001 |
| Iso. inv. Red sh. vs Green sh. | 660 to 1320 | Iso. inv.: 21 | Complete permutation N = 2097152 | P < 0.05 Except for 10 time-steps | P < 0.001 |
| Gr. cont. vs Gr. inv. | 0 to 600 | Gr. cont.: 5 vs Gr. inv.: 5 of 22 | Complete combination N = 26334 | P < 0.05 Except at time: 180 minutes | P = 0.53 |
| Gr. pooled Red sh. vs Green sh. | 0 to 600 | Gr. pooled: 27 | Complete permutation N = 134217728 | P < 0.05 | P < 0.0001 |
| Gr. cont. vs Gr. inv. Red sh. | 660 to 1320 | Gr. cont.: 5 vs Gr. inv.: 5 of 22 | Complete combination N = 26334 | P > 0.05 Except at time 210 | P = 0.023 |
| Gr. inv. Red sh. vs Green sh. | 660 to 1320 | Gr. inv.: 22 | Complete permutation N = 4194304 | P > 0.05 Except at time 1320 | P = 0.16 |
| Red sh. Iso. pooled Gr. pooled | 0 to 600 | Iso. pooled: 26 Gr. Pooled 26 of 27 | Random permutation N = $10^7$ | P < 0.05 Except for 7 time-steps | P < 0.001 |
| Red sh. Iso. pooled Gr. pooled | 660 to 1320 | Iso. pooled: 26 Gr. Pooled 26 of 27 | Random permutation N = $10^7$ | P > 0.05 Except for 4 time-steps | P = 0.66 |

Summary of permutation tests comparing shelter occupations (sh.) between isolated individuals (iso.) and groups (gr.) under different conditions: control condition (cont.), inversion condition (inv.) and Pooled conditions (control + inversion). The comparison between isolated individuals and groups is based on the proportion of individuals occupying the red shelter relative to the total population. The complete combination method is used to compare sheltered occupation between two conditions with the same population size (isolated *vs* isolated and groups *vs* groups). The complete permutation method is used to permute the population between two shelters for a set of trials. The random permutation method is used to compare the population in the red shelter between two conditions with different population sizes (isolated *vs* groups). For more details see Data analysis section of the Materials and Methods and the S1 File.

## Inversion condition and past events

For the inversion condition, the distribution of the difference between the population in the RG shelter and the GR shelter at 1320 minutes (last observation), shows that the isolated individuals are distributed among the shelters, with the majority inside the GR shelter and some outside (Fig 4). For the groups, the GR shelter is more frequently selected, and the distribution exhibits a bimodal pattern (Cramer-Von Mises test: T = 0.55, P < 0.0001, Estimated modes:

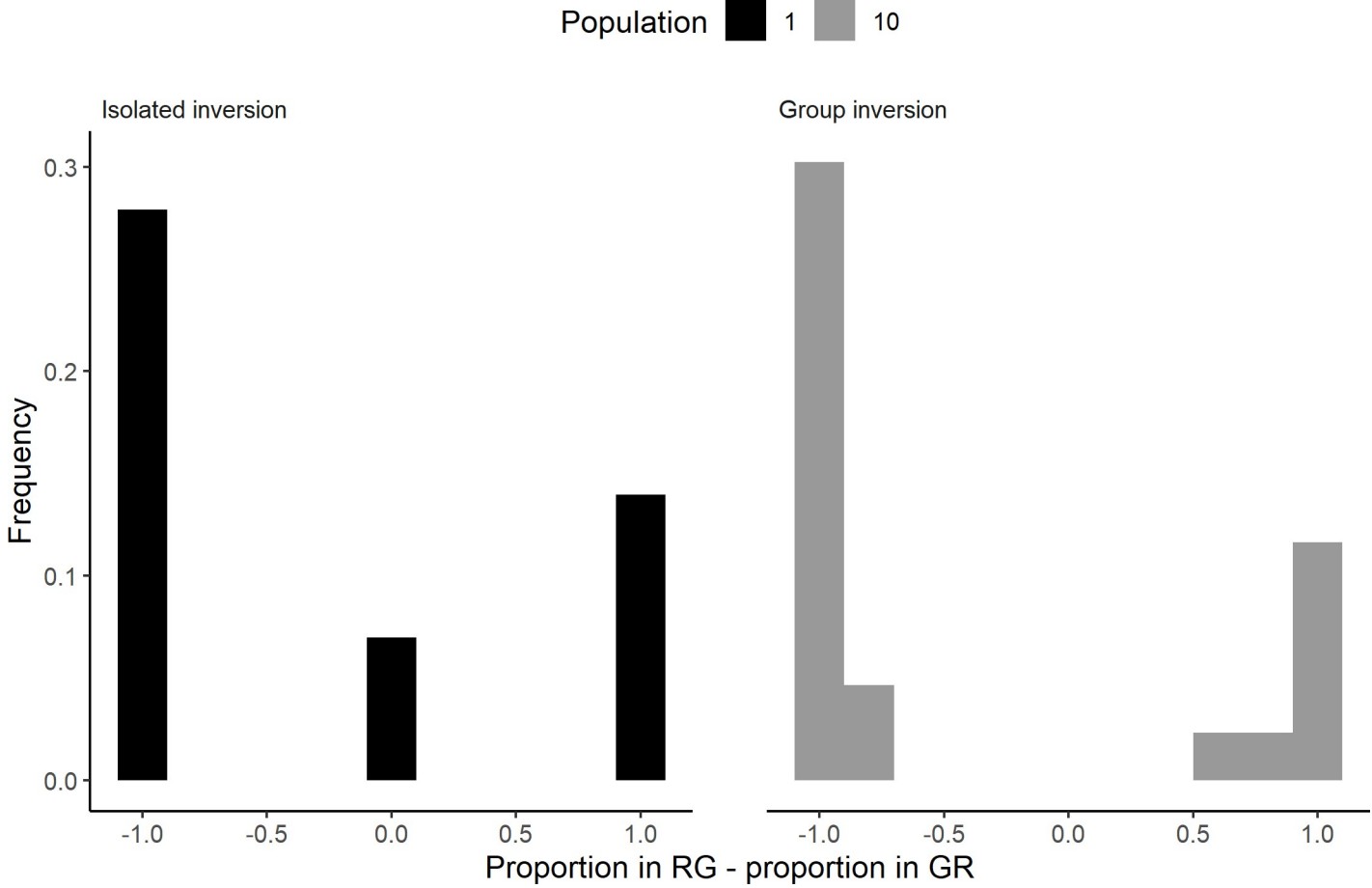

**Fig 4. Proportion of the total population in the RG shelter—GR shelter.** Distribution of the isolated individuals (black) and of the groups (grey).

-0.97/0.98 proportion of the total population, critical bandwidth = 0.37), indicative of interattractions. This bimodality becomes more and more prominent over time, starting from 660 minutes (Figs 5A and S5). Hence, consensus is reached relatively quickly after the inversion, implying a synchronized switch from the RG to the GR shelter (S6 Fig). At the end, all the groups reached a consensus, that is the significant majority of the sheltered individuals are in one shelter (GR or RG; binomial test: α = 0.05, and S1 Table Results section in S1 File; for more details see as well Materials and methods section).

These outcomes may be governed by the past events in the shelters (i.e. before the light inversion). Therefore, we focused mainly on the influence of the population inside the shelters at 600 minutes. For the isolated individuals, the presences in the RG shelter at 1320 minutes and at 600 minutes are independent ($X^2_{df = 1}$ = 0.611; P = 0.43; Table 2 and Fig 6). This suggests that individual memory or interindividual behaviours (e.g. idiosyncrasy) [62, 63] are negligible in our experiment. For the groups, the number of individuals in the RG shelter at the end of the trials and at 600 minutes are significantly correlated (Spearman test: $\rho$ = 0.58, P = 0.005, Fig 6). Furthermore, all individuals that stayed in the initially selected shelter are those that reached a consensus in the RG shelter at 600 minutes. In other words, they showed a consensual fidelity to this shelter ($F_{df = 1}$ = 6.1; P = 0.014; Table 2 and Fig 6). *A contrario*, all groups that did not reach a consensus at 600 minutes (in either shelter) did reached a consensus at 1320 minutes, but in the GR shelter (S5 and S6 Figs in Results section of S1 File).

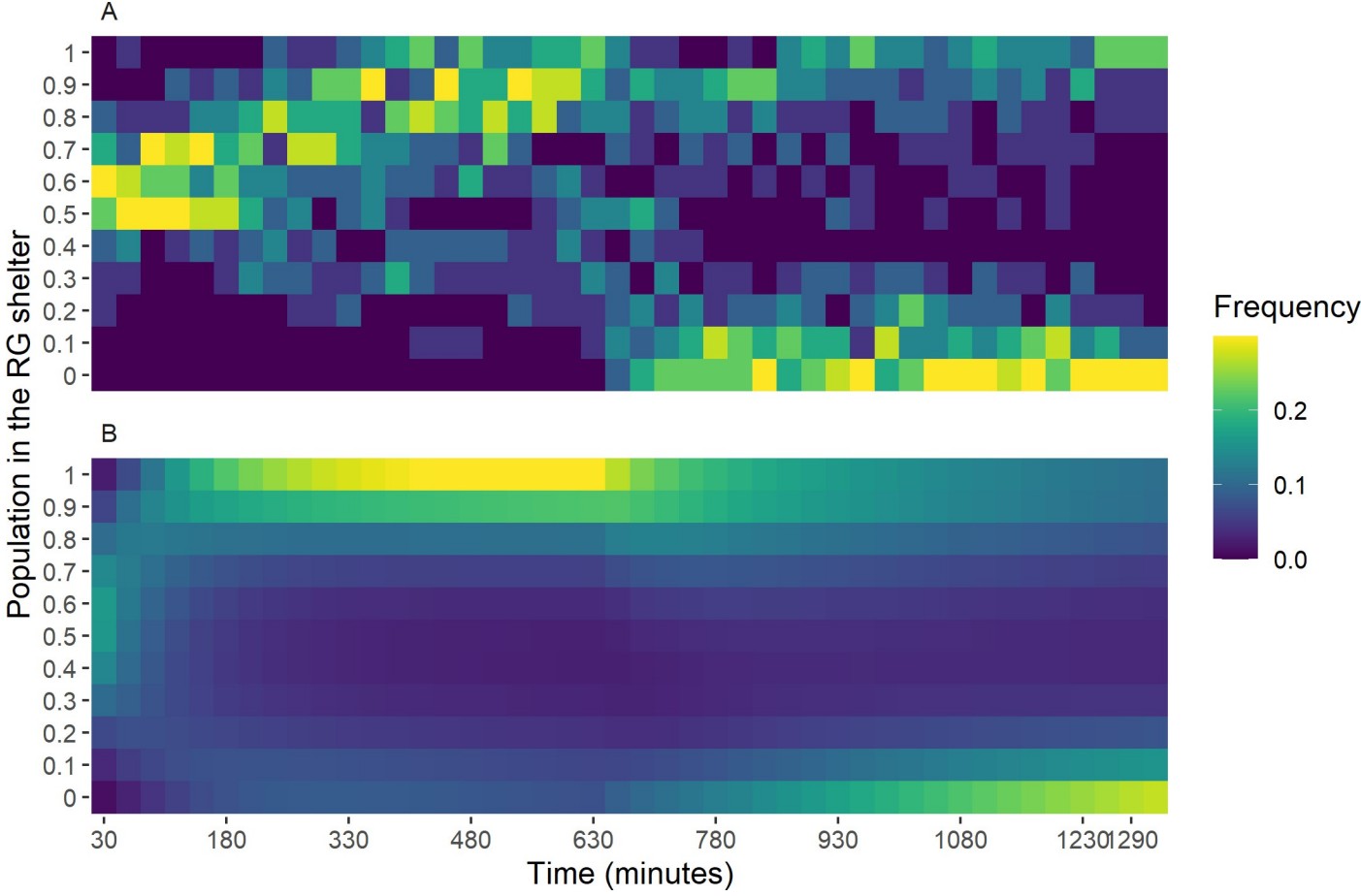

**Fig 5. Distribution of the population in the RG shelter.** A) Group inversion condition, from 30 to 1320 minutes. B) Theoretical group inversion condition, from 30 to 1320 minutes and at the stationary state (Inf), see eq. S6a-S6c in Model section in S1 File: parameters values: $\theta_r = 8 \times 10^{-4}$ s$^{-1}$, $\theta_g = 9 \times 10^{-4}$ s$^{-1}$, $\mu_r = 3.55 \times 10^{-3}$ s$^{-1}$, $\mu_g = 3.15 \times 10^{-3}$ s$^{-1}$ and $\xi = 0.302$.

**Table 2. Temporal dependence of the individual distribution.**

Isolated individuals

| 600 minutes ╲ 1320 minutes | Occupation inside the RG shelter | | Occupation outside the RG shelter | |
|---|---|---|---|---|
| | Observed | Expected | Observed | Expected |
| Occupation inside the RG shelter | 1 | 2.29 | 7 | 5.71 |
| Occupation outside the RG shelter | 5 | 3.72 | 8 | 9.28 |

Groups

| 600 minutes ╲ 1320 minutes | Groups > 7 individuals inside the RG | | Groups ≤ 7 individuals inside the RG | |
|---|---|---|---|---|
| | Observed | Expected | Observed | Expected |
| Groups > 7 individuals inside RG | 7 | 3.82 | 5 | 8.18 |
| Groups ≤ 7 individuals inside RG | 0 | 3.18 | 10 | 6.82 |

Observed and expected matrix of the occupation, at 600 minutes and at 1320 minutes, inside and outside the RG shelter for the isolated individuals and the occupation of more than seven individuals and less or equal to seven individuals inside the RG shelter for the groups. The expected matrix is obtained using the contingency table method [65]. The contingency table is used in a $X^2$ test for the isolated individuals and in a fisher test for the groups.

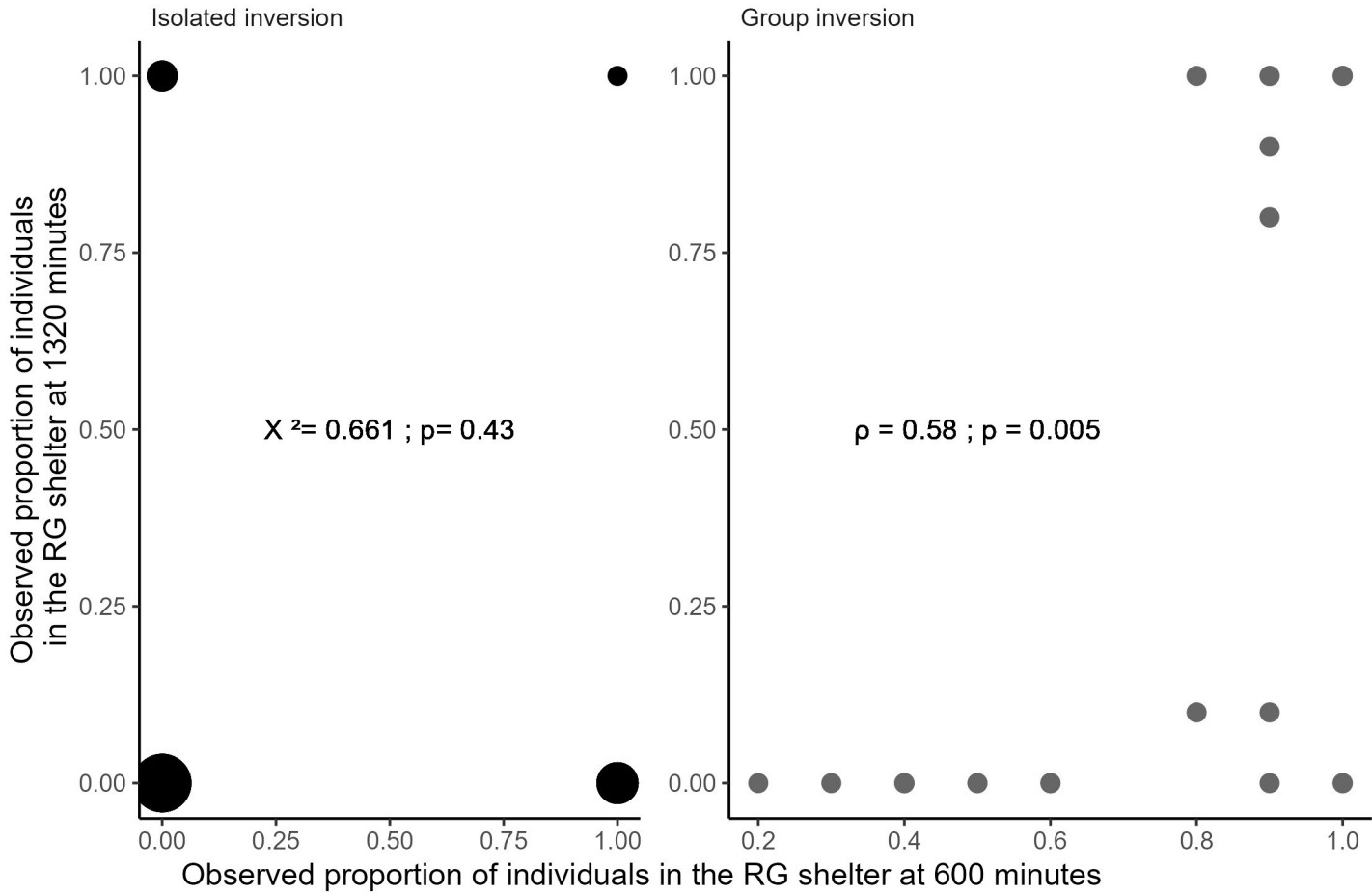

**Fig 6. Correlation chart between the proportion of the total population in the RG shelter at 600 minutes and at 1320 minutes.** For the isolated individuals (black) and for the groups (grey). Dot size indicate frequencies of observation.

These results suggest that the stability of the initial choice (before the light inversion) has a social component. After the inversion, the initial choice is only maintained for consensual groups and never for the isolated individuals or non-consensual groups. The inter-trials variability as seen by the bimodality displayed by the groups (Figs 4 and 5A), could be the result of various mechanisms involving inter-individual variabilities [64]. However in the sequel, we show, thanks to a mathematical model, that the interplay between individual behavioural stochasticity and interattraction is enough to fit the observed outcomes and to the system exhibit a collective memory [35, 37, 40].

## Model

We used a mathematical model validated by previous studies on the sheltering dynamics of *Blattella germanica* and *P. americana* [20, 53]. Its minimal hypotheses are:

1. All individuals behave in the same way, i.e. we neglected the idiosyncrasy [62];

2. The individual joining probability $\mu$ ($\mu_r$, $\mu_g$) for each shelter is constant and is larger for the red shelter;

3. The individual leaving probability of shelter depends on the maximal rate of leaving $\theta$ ($\theta_r$, $\theta_g$) and is larger for the green shelter;

4. The individual leaving probability decreases with the number of sheltered individuals, due to the interattractions $\xi$, which is independent of the light characteristics;

5. The $\theta$'s and the $\mu$'s are symmetrically inverted when the shelter luminosities are inverted. For more details see Model section in S1 File.

We analyse two complementary versions of the model. In the first one, we use a deterministic formulation (mean field equations: eqs. S3-S5, Model section in S1 File) to assess the collective behaviours generated by the model for different situations of shelter qualities. The second version, a master equation (eqs. S6, S7 in S1 File) is used to explore the role of stochasticity and to adjust the model parameters to the experimental conditions (Fig 5B). Both approaches highlight the importance of sociality (interplay between the strength of interattraction $\xi$ and the population size) and the role of shelter qualities (both relatively and absolute) in the observed phenomenon (S7–S10 Figs). For more details see Model section in S1 File.

## Before and after inversion dependencies

The model shows that the stability of the decision is crucially depending on the number of individual sharing a decision before the inversion. Fig 7 shows that the theoretical probability of observing more than 7 individuals in the RG at 1320 minutes increases with the proportion of individuals in the RG shelter at 600 minutes. This result is in agreement with our experimental results (Fig 6). Moreover, an exploration of the parameters of the model leads to the conclusion that group fidelity to the RG shelter is positively correlated to the strength of interattraction ($\xi$) (Fig 7). For small values of $\xi$ this distribution is monomodal while larger values of $\xi$ favour bimodal distributions (all or nothing responses), be it transiently (Fig 7) or at the stationarity (S10A Fig in Model section of the of S1 File). Not surprisingly, the difference between the shelter qualities influencing the rates of joining and leaving also modulate the distributions (S10B Fig in Model section of the S1 File).

## Discussion

Some groups having achieved a consensus in the red shelter before the light inversion showed a fidelity to this shelter (RG) while all groups that did not reach such consensus shifted to the new red shelter (GR). These collective responses, which are correlated to past events, emphasize the presence of a collective memory. From a theoretical point of view, the dynamics of the sheltering process are indicative of multiple stable states and potentially display hysteresis before and after a light inversion, as demonstrated by our model (S9 Fig). However, the selection of one of the stable states (consensus in RG or GR at the end of the experiment) does not only depend on the initial conditions (at t = 600 min), but also on the stochasticity. Indeed, due to the small number of individuals (N = 10) randoms events in the shelter (i.e., entries and exits) can lead some groups having reached a consensus in the RG shelter before the inversion to collectively shift towards the GR shelter. Our theoretical approach reveals that social interactions and the group's structure are sufficient to display a collective memory. Nonetheless, the robustness of such collective memory might be weaker than those involving individual memories to store the information. For example, it is well admitted that cultural transmission can be preserved over multiple generations [66, 67] thanks to the number or the proportion of informed individuals [68].

Despite its parsimony, the results of the model are in agreement with the observed patterns in our experiments (Fig 5A and 5B). It has been shown that in some situations individual

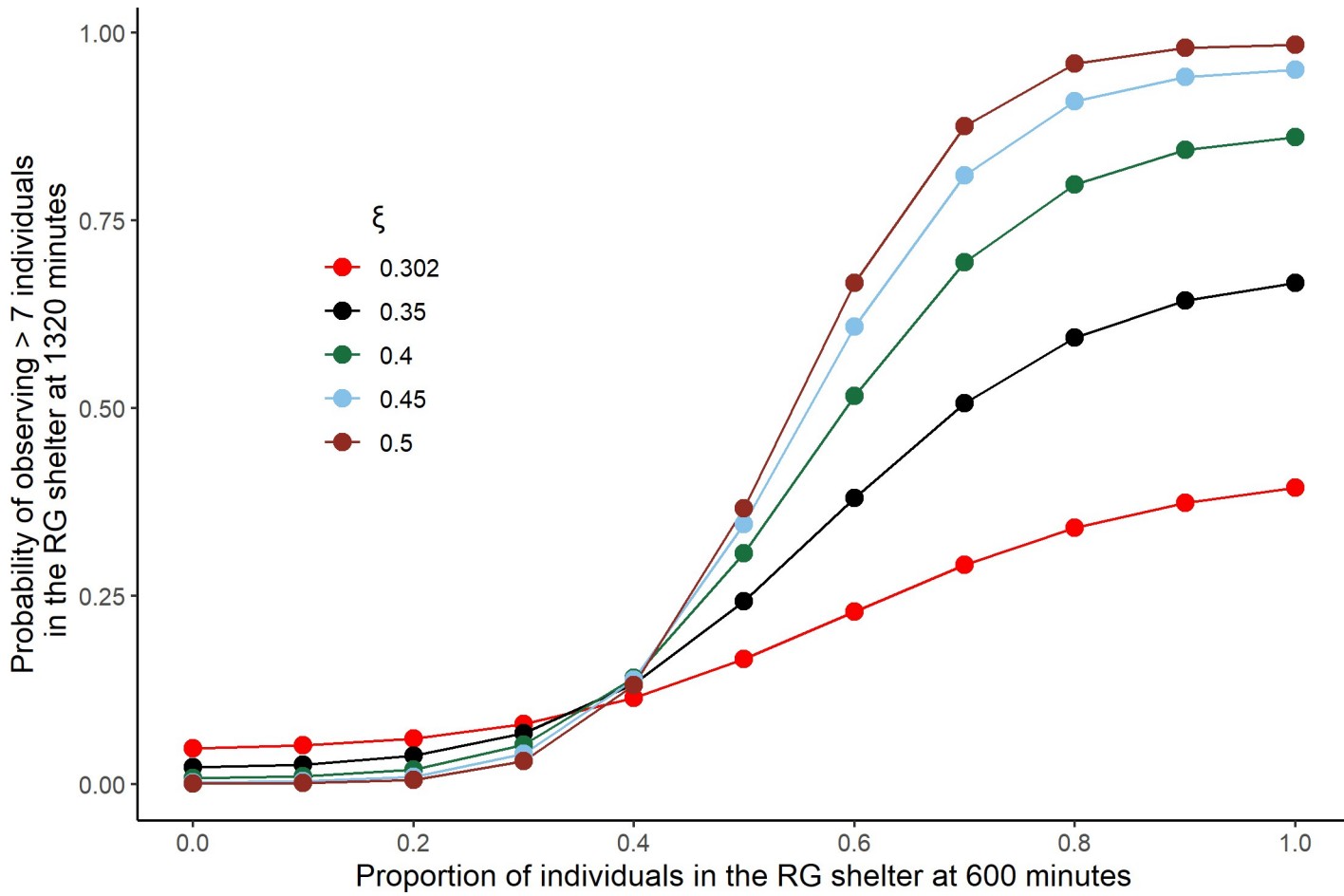

**Fig 7. Theoretical proportion of faithful groups for different values of interattractions ξ.** Probability of observing > 7 individuals in the RG shelter at 1320 minutes as function of the proportion of sheltered individuals in the same shelter at 600 minutes. Values obtain from eq. 6a-c (parameters values: N = 10; $\theta_r = 8\times10^{-4}$ s$^{-1}$, $\theta_g = 9\times10^{-4}$ s$^{-1}$, $\mu_r = 3.55\times10^{-3}$ s$^{-1}$, $\mu_g = 3.15\times10^{-3}$ s$^{-1}$).

memory plays a role in the spatio-temporal distributions of individuals [37, 40]. In our case, such individual capability is not observed, although several studies highlighted that cockroaches are able to learn and retain new information [69, 70]. Other mechanisms can also be at work and buffer some effects of the imposed perturbation. For example, hydrocarbons marking contributes to the stabilisation of the aggregate [50] and plays the role of more perennial external storage for the collective memory.

The observed difference of outcomes between isolated individuals and aggregated individuals can be attributed to several factors, including the fitness consequences of aggregation, of the qualities of the selected site and environmental explorations. Despite the risks involved in exploring new environment, leaving a degraded site enables isolated individuals, or small groups to locate a more suitable shelter or join a large cluster that offers cooperative benefits [71, 72]. In the case of larger groups displaying fidelity to the degraded site, the cooperative gained benefits may overcome the losses due to the site degradation and the risks associated with exploration.

Similar networks of feedbacks than those highlighted in this paper have been described to be present in other social systems [22, 73, 74]. A body of works devoted to such social systems where positive feedbacks are in competition highlights the emergence of multistable patterns

of organization that depends on different elements such as the size of the system, the environmental parameters and the strength of the interactions/interattractions [58, 75]. Another ingredient playing a major role in the dynamics of these systems is the stochasticity. While these elements may have different weights depending on the species studied, the collective capabilities and in particular the collective memory of the patterns adopted before an imposed environmental change on the system -revealed by the present study–are expected to be present in a wide range of systems, from animal behaviour to artificial dynamics. The values of the parameters quantifying the behaviour of individuals makes it possible to easily explain similarities between species or differences between groups of the same species, these values resulting from multiple biological causes (the physiological state of individuals, their kinship [76] or an environmental influence [12]).

## Supporting information

**S1 Fig. Experimental set-up.**
(TIF)

**S2 Fig. Camera position and filming zone.**
(TIF)

**S3 Fig. Sheltering process of the total population over time.** Mean ± SEM of isolated individuals (black) and of groups (grey).
(TIF)

**S4 Fig. Sheltered population over time.**
(TIF)

**S5 Fig. Group inversion condition: Distribution of individuals between the shelter (XRG—XGR) for every time-step.**
(TIF)

**S6 Fig. Proportion of the total population sheltered in the RG shelter (square) and in the GR shelter (cross) over-time.** A-G) Consensual groups in the RG at 600 and at 1320 minutes.
(TIF)

**S7 Fig. Bifurcation diagrams of the stationary solutions of the eq. S5 in S1 File as a function of $\zeta$.**
(TIF)

**S8 Fig. Number of stable solutions given by eq. S5 in S1 File as a function of gamma and zeta.**
(TIF)

**S9 Fig. Bifurcation diagram of eq. S5 in S1 File.**
(TIF)

**S10 Fig. Sheltered population in the RG shelter at the steady state.** Integration from eq. S6a-S6c in S1 File.
(TIF)

**S1 File.**
(DOCX)

## Author Contributions

**Conceptualization:** Mariano Calvo Martín, Esméralda Rodriguez Palacio, Jean-Louis Deneubourg, Stamatios C. Nicolis.

**Data curation:** Esméralda Rodriguez Palacio.

**Formal analysis:** Mariano Calvo Martín, Esméralda Rodriguez Palacio, Jean-Louis Deneubourg, Stamatios C. Nicolis.

**Supervision:** Stamatios C. Nicolis.

**Visualization:** Mariano Calvo Martín.

**Writing – original draft:** Mariano Calvo Martín, Jean-Louis Deneubourg, Stamatios C. Nicolis.

**Writing – review & editing:** Mariano Calvo Martín, Esméralda Rodriguez Palacio, Jean-Louis Deneubourg, Stamatios C. Nicolis.

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
