## [Decision Letter · Decision Letter 0]

2 Feb 2023

PONE-D-22-34285Emergence and retention of a collective memory in cockroachesPLOS ONE

Dear Dr. Calvo Martín,

Thank you for submitting your manuscript to PLOS ONE. After careful consideration, we feel that it has merit but does not fully meet PLOS ONE’s publication criteria as it currently stands. Therefore, we invite you to submit a revised version of the manuscript that addresses the points raised during the review process.

The experiments presented in this research have been conducted rigorously, with appropriate controls and replication. The sample sizes are large enough to produce robust results and methods and reagents are described in sufficient detail for another researcher to reproduce the experiments described. The data presented in the results section of the manuscript supports the conclusions drawn and the research meets all applicable standards for the ethics of experimentation and research integrity.

After a lengthy process of recruiting reviewers, we have obtained a single review, hence I highly recommend following the suggestions given by the reviewer and resubmitting your paper after major revision.

We look forward to receiving your revised manuscript.

Kind regards,

Lemana Spahić

Academic Editor

PLOS ONE

Reviewers' comments:

Reviewer's Responses to Questions

**Comments to the Author**

1. Is the manuscript technically sound, and do the data support the conclusions?

Reviewer #1: Yes

2. Has the statistical analysis been performed appropriately and rigorously? 

Reviewer #1: Yes

3. Have the authors made all data underlying the findings in their manuscript fully available?

Reviewer #1: No

4. Is the manuscript presented in an intelligible fashion and written in standard English?

Reviewer #1: No

5. Review Comments to the Author

Reviewer #1: In this study, the authors perform experimentation with cockroaches to discern their decision making processes when faced with two choices of light conditions. One of the choices (red light) is apparently favoured by cockroaches over the green light which is also what the authors find. However, when applied with an inversion of the light conditions, the findings differ between individual and group behaviours. The authors find that while the (isolated) individuals can transition from the less preferred zone to the preferred zone, the groups show inability to do that. Hence, existence of collective memory is suggested whereby group choice or ‘inertia’ prevents individuals to pursue their choices. The results are explained by a mathematical model (with two versions: deterministic and stochastic). The deterministic model is a differential equation model with the rate of change of number of individuals in a particular zone being modeled by the number of free or available individuals that can move to that zone, individual joining, and leaving rates. The leaving rate, in the model, is multiplied by an exponentially decreasing function of the number of individuals in that zone. This, supposedly, captures the trend observed in the experiments; for small group size the model shows flexibility in decision making, i.e. the proportion of individuals in a given zone decreases continuously if the individuals do not prefer that zone. On the other hand, for large group sizes, the proportion decreases, however, in a discontinuous manner or shows hysteresis. While I like the overall idea of both the experiments and the model, I have some criticisms and I state them as major comments.

Major comments.

• The choice of the exponentially decaying function with number of individuals in a zone that modulates the leaving rate is not justified enough. For a reader to understand the origin of this choice is crucial not only to appreciate the modelling part but to link it with the actual behaviours. Hence, I expect more justification by adding a paragraph on it. It took me some time and also I had to cross check the reference cited (Calvo

Mart´ın et al. 2019) to understand the motivation behind the choice. While, I mechanistically understand, how this choice can lead to the observed experimental results, it is yet unclear to me why this choice was made. For instance, a modulation of the joining rate with a similar function that increases exponentially with the number of individuals in that zone may also address the observation. (The authors claim that they find no evidence for attraction in their experiments, but it was unclear to me what is the basis of this claim.) Or, for that matter, both the modulation functions can depend on the number of individuals in a zone, however their relative strengths can be different. Hence, I think that this point could be addressed by making the model more general to modulate both the joining as well as leaving rate. More broadly, alternative models (at least one) and why they may as well or may not work should be checked and included in the text if possible. If the modelling is not possible, a small survey of literature of similar models should be incorporated.

• The model does show hysteresis as a function of the individual choice for large group size. However, since this particular result hasn’t been shown in the experiments, the claim particularly about hysteresis should be toned down. A possible experimentation to show this could be inverting the light conditions in a more gradual or continuous manner. Two initial conditions for different group sizes might be used. One: keeping all individuals in the preferred zone at the start, and two: keeping all of them in the non preferred zone. A

plot of the proportion of individuals in either of the zone as a function of changing lighting conditions might actually be able to show hysteresis properly in experiments. However, this, I understand, is definitely out of the current scope of the work. Therefore, I suggest to avoid using hysteresis. The authors have right now shown the existence of collective memory or inertia but just for an abrupt switch of the conditions which I think is insufficient to prove hysteresis.

• The use of the stochastic version of the model was not clear to me. Basically, if the deterministic version is capable of capturing the observed trends, then the reasons to make it stochastic are not clear.

Minor comments.

I found several grammatical mistakes and also typos (especially in equations) that must be fixed. Some of the following points are also not really minor as they deal with the readability of the paper and are, therefore, very important for building an understanding while reading the text; While I think the experimentation and the ideas are not that difficult to grasp, overall, I found it very hard to follow them through the text. Therefore, the writing needs considerable work. I only state few of the points hoping that everything gets rectified through a thorough revision by the authors.

• Line 162: Definition of sheltered proportion missing.

• Line 168: Interpretation missing – What is meant by weak effect of time? Can it be written more simply? No figure is referred to highlight this effect.

• Line 191: Definition and unit of cumulative occupation?

• Line 192: Claim of difference between group choice and individual is unclear until the order parameter is defined clearly at the start.

• Line 195 - 198: Reference missing.

• Line 198: What thresholds are we talking about here? The authors cannot leave it for the readers to figure out themselves.

• In so many place use of comma is entirely missing. Just for an example, in line 205, the sentence should be – Indeed, if the quality of ... decreases, the decision ... (Note the use of comma).

• Line 218: Isn’t the interpretation simple: that the groups are unable to respond quickly or react to environ- mental changes?

• Fig 6 is referred before figure 5 in the main text. Hence, change their numbering order.

• Line 227: The plot seems unimodal to me even after 660 minutes.

• Definition of consensual fiedelity not clear

• Supplementary text line 121: isn’t it attraction of unsheltered to sheltered?

• Supplementary text line 134: It should be mentioned that the solution is for a steady state by putting derivatives equal to zero.

• Supplementary text line 206: Typos in equation.

6. PLOS authors have the option to publish the peer review history of their article (what does this mean?). If published, this will include your full peer review and any attached files.

Reviewer #1: No

---

## [Author Response · Author response to Decision Letter 0]

31 Mar 2023

Please see the new revised version of the manuscript and the point by point response to the reviewer comments.

---

## [Decision Letter · Decision Letter 1]

30 May 2023

PONE-D-22-34285R1Emergence and retention of a collective memory in cockroachesPLOS ONE

Dear Dr. Calvo Martín,

Thank you for submitting your manuscript to PLOS ONE. After careful consideration, we feel that it has merit but does not fully meet PLOS ONE’s publication criteria as it currently stands. Therefore, we invite you to submit a revised version of the manuscript that addresses the points raised during the review process.

The decision is “Minor Revision” to give you a chance to respond to the comments of Reviewer #1. If this is done adequately, I will most likely be able to accept the manuscript without involving reviewers again.

We look forward to receiving your revised manuscript.

Kind regards,

Wolfgang Blenau

Academic Editor

PLOS ONE

Journal Requirements:

Reviewers' comments:

Reviewer's Responses to Questions

**Comments to the Author**

1. If the authors have adequately addressed your comments raised in a previous round of review and you feel that this manuscript is now acceptable for publication, you may indicate that here to bypass the “Comments to the Author” section, enter your conflict of interest statement in the “Confidential to Editor” section, and submit your "Accept" recommendation.

Reviewer #1: All comments have been addressed

2. Is the manuscript technically sound, and do the data support the conclusions?

Reviewer #1: Yes

3. Has the statistical analysis been performed appropriately and rigorously? 

Reviewer #1: I Don't Know

4. Have the authors made all data underlying the findings in their manuscript fully available?

Reviewer #1: No

5. Is the manuscript presented in an intelligible fashion and written in standard English?

Reviewer #1: Yes

6. Review Comments to the Author

Reviewer #1: I am satisfied with the author's response and changes made to the manuscipt and the supplemental material. However, minor grammatical errors and typos still remain. I would lke to suggest that the captions of Table 1 and 2 in the main text can be expanded that will help the readers to understand the technical analysis performed there. Finally, I could hardly understand Figure 6 of the main text due to its poor quality. Apart from these concerns, tha manuscript seems good for a publication. The conclusions about collective memory inferred through intelligent experiments and a rigorous mathematical model are very interesting and should be welcomed by the community of researchers in this area.

7. PLOS authors have the option to publish the peer review history of their article (what does this mean?). If published, this will include your full peer review and any attached files.

Reviewer #1: No

---

## [Author Response · Author response to Decision Letter 1]

9 Jun 2023

Please see "Reviewer comments", in there we address the reviewer concerns

---

## [Editor Report · Decision Letter 2]

13 Jun 2023

Emergence and retention of a collective memory in cockroaches

PONE-D-22-34285R2

Dear Dr. Calvo Martín,

We’re pleased to inform you that your manuscript has been judged scientifically suitable for publication and will be formally accepted for publication once it meets all outstanding technical requirements.

Kind regards,

Wolfgang Blenau

Academic Editor

PLOS ONE
---

## [Editor Report · Acceptance letter]

23 Jun 2023

PONE-D-22-34285R2 

Emergence and retention of a collective memory in cockroaches 

Dear Dr. Calvo Martín:

I'm pleased to inform you that your manuscript has been deemed suitable for publication in PLOS ONE. Congratulations! Your manuscript is now with our production department. 

Kind regards, 

on behalf of

Dr. Wolfgang Blenau 

Academic Editor

PLOS ONE